# Under-Reporting Cases and Deaths from Melioidosis: A Retrospective Finding in Songkhla and Phatthalung Province of Southern Thailand, 2014–2020

**DOI:** 10.3390/tropicalmed8050286

**Published:** 2023-05-20

**Authors:** Jedsada Kaewrakmuk, Sarunyou Chusri, Thanaporn Hortiwakul, Soontara Kawila, Wichien Patungkaro, Benjamas Jariyapradub, Pattamas Limvorapan, Bongkoch Chiewchanyont, Hathairat Thananchai, Kwanjit Duangsonk, Apichai Tuanyok

**Affiliations:** 1Faculty of Medical Technology, Prince of Songkla University, Hatyai, Songkhla 90110, Thailand; jedsada.k@psu.ac.th; 2Faculty of Medicine, Chiang Mai University, Chiang Mai 50200, Thailand; hathairat.t@cmu.ac.th (H.T.);; 3Faculty of Medicine, Prince of Songkla University, Hatyai, Songkhla 90110, Thailand; sarunyouchusri@hotmail.com (S.C.); hratri@medicine.psu.ac.th (T.H.); soontara_aom@hotmail.com (S.K.); 4Hatyai Hospital, Hatyai, Songkhla 90110, Thailand; vichan2509@gmail.com; 5Songkhla Provincial Hospital, Songkhla 90000, Thailand; bactsk@hotmail.com; 6Phatthalung Provincial Hospital, Phatthalung 93000, Thailand; bo.patta@hotmail.com; 7The Office of Disease Prevention Control 12, Songkhla 90000, Thailand; 8Department of Infectious Diseases and Immunology, College of Veterinary Medicine, University of Florida, Gainesville, FL 32608, USA

**Keywords:** melioidosis, *Burkholderia pseudomallei*, southern Thailand

## Abstract

Melioidosis, caused by *Burkholderia pseudomallei*, is a notifiable disease associated with a high mortality rate in Thailand. The disease is highly endemic in northeast Thailand, while its prevalence in other parts of the country is poorly documented. This study aimed at improving the surveillance system for melioidosis in southern Thailand, where the disease was believed to be underreported. Two adjacent southern provinces, Songkhla and Phatthalung, were selected as the model provinces to study melioidosis. There were 473 individuals diagnosed with culture-confirmed melioidosis by clinical microbiology laboratories at four tertiary care hospitals in both provinces from January 2014 to December 2020. The median age was 54 years (IQR 41.5–64), 284 (60%) of the patients were adults ≥50 years of age, and 337 (71.2%) were male. We retrospectively analyzed 455 patients treated at either Songklanarind Hospital, Hatyai Hospital, Songkhla Provincial Hospital, or Phatthalung Provincial Hospital, of whom 181 (39.8%) patients died. The median duration from admission to death was five days (IQR 2–17). Of the 455 patients, 272 (57.5%) had at least one clinical risk factor, and 188 (39.8%) had diabetes. Two major clinical manifestations, bacteremia and pneumonia, occurred in 274 (58.1%) and 166 (35.2%) patients, respectively. In most cases, 298 (75%) out of 395 local patients were associated with rainfall. Over the seven years of the study, the average annual incidence was 2.87 cases per 100,000 population (95% CI, 2.10 to 3.64). This study has confirmed that these two provinces of southern Thailand are endemic to melioidosis; even though the incidence rate is much lower than that of the Northeast, the mortality rate is comparably high.

## 1. Introduction

The global burden of melioidosis has been estimated to be 165,000 human cases per year, with 89,000 deaths [1]. This estimation has also suggested that melioidosis is severely underreported in 45 countries, including Thailand, where it is known to be highly endemic. Melioidosis is caused when the bacterium *B. pseudomallei* enters a person’s body through cuts or skin lesions, inhalation of contaminated dust, or consumption of contaminated food or water. Without treatment, the mortality rate of this disease approaches 90%, but with a rapid response and proper antibiotics, this can be reduced to 40% or less [2]. A recent study from Thailand has indicated that melioidosis is a significant cause of death and that its clinical epidemiology differs between regions [3]. There were 7126 cases reported from 60 hospitals countrywide from 2012 to 2015. The most incidences were from the Northeast with 5475 cases, while the most undersized majority was from the West with only 19 cases. Other regions, including the Central, the South, the North, and the East, had 536, 374, 364, and 358 cases, respectively [3]. This may suggest that the annual incidence of melioidosis in Thailand is underreported, at least four times below the estimated 7572 cases by modeling. In addition, this recent study has also shown that only 126 (4%) deaths were reported to the national notifiable diseases surveillance system in that 4-year period, which was too low compared with the estimated 2838 deaths annually by the modeling [1]. Furthermore, in the review by Hinjoy and colleagues in 2018, the mortality rate of melioidosis throughout Thailand was approximately 35% [4]. This mortality rate was likely calculated solely based on reports from hospitals in the Northeast. Therefore, it has been questioned whether melioidosis is endemic in other parts of Thailand, especially in the south, which has a border with Malaysia, where melioidosis is reported frequently.

*B. pseudomallei* has been discovered in the environment of southern Thailand by multiple studies, including the studies by Finkelstein and colleagues in the 1960s [5], a following study by Nachiangmai and colleagues in the 1980s [6], and then as part of a soil survey campaign throughout Thailand by Vuddhakul and colleagues in the 1990s [7]. The endemicity of melioidosis in southern Thailand became known when foreign tourists developed melioidosis following the 2004 tsunami. Some were hospitalized in Thailand [8], while others developed the disease when they returned to their home countries [9,10]. Another major outbreak of melioidosis occurred in Koh Phangan, an island in the Gulf of Thailand, when 11 individuals, including three foreign tourists and eight local villagers, contracted melioidosis from January to March 2012, three of whom died, one of whom was a neonate [11]. *B. pseudomallei* was found in water sources throughout the island, and these could cause infections even though the genotypes of the isolates from patients and water were different. Two retrospective studies of melioidosis have been conducted in the region, both in Songkhla Province, the second-largest province in southern Thailand. The first study reported 59 pediatric melioidosis cases admitted to Hatyai Hospital from 1959 to 1989, with 81% fatalities among patients with septicemia and 100% in neonates [12]. The significant finding of that study was a very high mortality rate in pediatric melioidosis. The second study reported 134 culture-confirmed cases treated at Songklanakarind Hospital with 8.9% mortality from 2002 to 2011 [13]. With the lower incidences compared with those from the Northeast, healthcare professionals always consider melioidosis an uncommon disease in southern Thailand.

Notably, based on the findings from the 7th World Melioidosis Congress (WMC) held in Thailand in 2013, two clinical needs for melioidosis were pressing: one was to reduce the global incidence of the disease, and another was to improve the clinical outcome of infected patients [14]. For Thailand, we firmly believe that without a proper surveillance system, we would be unable to identify the actual burden of melioidosis, while the estimated number of cases and deaths is relatively high. Since 2014, we have established an initiative for melioidosis in southern Thailand with local universities, hospitals, and health authorities. Herein, we report the recent findings from our retrospective study of melioidosis conducted in two southern provinces, Songkhla and Phatthalung, that included data obtained from clinical microbiology laboratories and patients’ records at four tertiary care hospitals. We can confirm that melioidosis in southern Thailand is seasonal with a low incidence rate, while the mortality rate is comparable with the data from the Northeast.

## 2. Materials and Methods

### 2.1. Study Design, Sites, and Data Collection

Information on culture-confirmed melioidosis cases was collected from clinical microbiology laboratories at four tertiary-care hospitals, including Songklanakarind Hospital, Hatyai Hospital, Songkhla Provincial Hospital, and Phatthalung Hospital, from January 2014 to December 2020. Culture-confirmed melioidosis was defined as a patient with a culture positive for *B. pseudomallei* grown from any clinical specimen site. Techniques for *B. pseudomallei* identification were based on a standard hospital guideline, including a series of biochemical tests and/or the VITEK-2 Compact system. The bacterial isolate(s) from each tested specimen was collected, further processed in the BSL-2+ laboratory, and confirmed as *B. pseudomallei* by real-time PCR assays targeting TTS-1, BTFC, and YLF species-specific genes [15,16].

### 2.2. Data Sets and Analyses

Relevant microbiological data were collected from the hospital laboratory database, while information on patients was collected from hospital records. A questionnaire was used to record patients’ data, including their demographic data (e.g., age, gender, occupation, and household location), any current and/or previously diagnosed underlying diseases (e.g., diabetes, cancer, renal disease, thalassemia, excessive alcohol consumption, other lung diseases, HIV, and immunosuppressant use), significant clinical presentations, and outcome. The data presented in this study were analyzed based on 455 cases of patients who were admitted and received treatments in one of these four study hospitals. The case fatality rate (CFR) was calculated based on the proportion of patients who died from melioidosis among all patients presenting with the same clinical manifestations, or the same culture-positive specimens, over the seven years of the study, and then multiplied by 100 to yield a percentage.

The incidence rates and the number of cases associated with rainfall amounts were calculated based on 395 patients who were residents of Songkhla or Phatthalung Province. The average rainfall was calculated based on the data provided by multiple meteorological stations in both provinces. The average incidence rate per 100,000 population per year was calculated by combining the total population of the Songkhla and Phatthalung provinces. The average incidence rate of melioidosis in each district was also calculated based on the number of cases per 100,000 population of each district per year and then mapped using the program QGIS 3.26.2 ‘Buenos Aires’, a free and open-source geographic information system (https://qgis.org). The district map of Songkhla and Phatthalung Provinces was created by the GEO-Informatic Research Center at Prince of Songkla University, Thailand. 

### 2.3. Statistical Analysis

A median with an interquartile range (IQR) was used for a non-normal distribution. When estimating a population parameter, e.g., mean/average incidence rate, a 95% confidence interval was computed to get a sense of the estimate’s precision. A percentage was used to determine the frequency for each data category. The Pearson correlation coefficient (r) was calculated to determine the strength and direction of the relationship between the amount of rainfall (*x*) and the number of monthly melioidosis cases (*y*), where *n* is the number of data pairs, using the equation below.
$$r=\frac{n(\sum xy)-(\sum x)(\sum y)}{\sqrt{[n\sum x^{2}-\left(\sum x{)}^{2}\right][n\sum y^{2}-\left(\sum y{)}^{2}\right]}}$$

### 2.4. Ethical Consideration

Ethical permissions for this study were obtained from the Ethical Committee of the Faculty of Medicine at Prince of Songkla University (No. 59-370-14-1) and the Research Ethic Committee of Hatyai Hospital (No. 52/2561). 

## 3. Results

### 3.1. Diagnostic Specimens

A total of 1013 diagnosed clinical specimens collected from 473 patients (455 inpatients and 18 outpatients) grew *B. pseudomallei* based on the laboratory records at four study hospitals from January 2014 to December 2020. Hemocultures were the most common positive specimens found in 324 (71.2%) inpatients, followed by pus and wound swabs (120, 26.4%), sputa (104, 22.9%), urines (32, 7.0%), synovial fluids (20, 4.4%), parotid gland abscesses (18, 4.0%), plural fluids (6, 1.3%), spleen and liver abscesses (6, 1.3%), and cerebrospinal fluids (CSF, 2, 0.4%). We noted that at least 117 (25.7%) inpatients had positive samples from multiple sites. Details are in Table 1.

### 3.2. Demographic and Clinical Characteristics of Patients

Of the 473 patients, 455 were inpatients admitted for treatments, while 18 were outpatients whose clinical specimens were collected and cultured positive without being followed up for treatments at any of these four study hospitals. Clinical data from these 18 outpatients were incomplete and not used in the retrospective analysis. There were 472 melioidosis episodes associated with these 455 inpatients. Fifteen (3.3%) patients were admitted more than once, 11 of whom were readmitted after six months of their first admissions. The other four patients were readmitted during the eradication phase (taking oral treatments at home). Two patients were readmitted three times within 2 and 4 years, respectively. In addition, 10 admitted patients at Songklanakarind Hospital, a hospital affiliated with the medical school, were referred from the other three hospitals. Therefore, these patients were diagnosed twice, and the number of episodes was counted at both locations. The number of culture-confirmed melioidosis episodes diagnosed at each study hospital in each year is summarized in Table 2.

Most melioidosis patients were male (337 cases, 71.2%). The median age was 54 years (IQR 41.5–64), and 284 (60%) of the patients were adults ≥ 50 years of age. At least 99 (21%) of the patients are farmers or have occupations (e.g., construction workers, park rangers) involving soil/water contact daily. Based on the clinical data collected from 455 inpatients, the majority of them (272, 59.8%) had at least one underlying disease or condition, including diabetes mellitus (188, 41.3%), cancer or malignancy (35, 7.7%), chronic renal diseases (26, 5.7%), immunosuppressant usage (24, 5.3%), thalassemia (22, 4.8%), excessive alcohol consumption (14, 3.1%), chronic lung diseases (14, 3.1%), and HBV/HIV (13, 2.9%). Bacteremia was one of the most common clinical manifestations found in 274 (58.1%) out of 472 episodes, followed by pneumonia (166, 35.2%), skin or wound infection (101, 21.4%), urinary tract infection (46, 9.7%), hepatosplenic abscess (33, 7.0%), septic arthritis (18, 3.8%), parotid gland infection (18, 3.8%), and central nervous system involvements (2, 0.4%). 

Deaths from melioidosis occurred rapidly. One hundred and eighty-one, or 39.8%, of the treated patients (inpatients) died, of which 56 (30.9%) deaths occurred within the first two days of their admissions, 46 (25.4%) died between days 3 and 7, and the other 79 (43.7%) died after 7 days of their admissions. We also noted that the case fatality rate (CFR) was also high in patients presenting with pneumonia (56.7%), urinary tract infection (46.7%), or bacteremia (45.8%) (Table 3). Moreover, patients diagnosed with culture-positive results from sputum, hemoculture, urine, or multiple specimens also had high CFRs (Table 1). 

### 3.3. Geographic Distribution of the Patients

Based on the hospital records, 390 (82.5%) out of 473 melioidosis patients were local residents of Songkhla or Phatthalung, while 70 (14.8%) patients were referrals from hospitals in other southern provinces. The other 13 (2.7%) patients were non-Southern Thailand residents. The average incidence rate of melioidosis in Songkhla and Phatthalung combined was 2.87 cases per 100,000 population (95% CI, 2.10 to 3.64) per year, which was calculated based on the 390 cases of the local residents. The incidence rates varied from 0.00 to 6.23 cases per 100,000 population per year among 27 districts (16 in Songkhla and 11 in Phatthalung). Na Mom District of Songkhla Province had the highest average incidence rate at 6.23 cases per 100,000 population per year, while Krasaesin District of the same province had no melioidosis cases (Figure 1).

### 3.4. Melioidosis Cases Correlated with Rainfall

Seasonally, southern Thailand is affected by two monsoons: one from the Indian Ocean, usually at the beginning of May, and then continues with another from the Pacific Ocean, starting in late October and lasting until December. Generally, both provinces have a long rainy season with eight months of rain. In this study, 298 (76.4%) out of the 390 local cases occurred during the rainy season. Especially in December, when it sometimes had flooding; as a result, the number of melioidosis cases increased in January of the following year. We noted that the increased number of melioidosis cases in 2016–2017 was associated with two major flooding events that occurred at the end of both years (Figure 2a). Based on the Pearson correlation coefficient (r) analysis, we have found that the increased number of melioidosis cases in the rainy season had the strength of a linear association with the amount of rainfall. The r value was determined at 0.54 or 0.64 when the number of melioidosis cases in the same month of the rainfall or from the following month was used in the calculation, respectively (Figure 2b,c). 

## 4. Discussion

We were enthused by the finding from the 7th WMC held in Bangkok, Thailand, in 2013 that suggested two urgent clinical needs for melioidosis: one was to reduce the global incidence of the disease, and another was to improve the clinical outcome of infected patients [14]. Other than finding the actual burden of melioidosis in any endemic areas, knowing the extent of the problems will also help identify the needed scale of the solution to fulfill these two urgent needs. Hence, this investigation was initiated based on the hypothesis that melioidosis in southern Thailand was endemic but underreported. We chose two adjacent provinces, Songkhla and Phatthalung, to study melioidosis because (i) four tertiary-care hospitals served almost two million people living within 4177 mi^2^ of both provinces and (ii) the laboratorians and clinicians at these hospitals had experience in diagnosing and treating melioidosis based on their previous reports [12,13]. It is worth mentioning that Songklanakarind Hospital, Hatyai Hospital, and Songkhla Provincial Hospital have 1000, 700, and 508 beds, respectively, and all are located in Songkhla Province, while Phatthalung Provincial Hospital is the only tertiary-care hospital with 450 beds in Phatthalung Province, located approximately 62 miles away from the other three. Based on the most recent hospital statistical data, these were 250,235, 58,484, 36,101, and 36,345 inpatients admitted for treatments at these four hospitals in 2022, respectively. Songklanakarind Hospital is the largest hospital in the south and is affiliated with a medical school and a referral medical center for southern Thailand. The major findings from this investigation are discussed below.

### 4.1. Melioidosis in Southern Thailand Is Seasonal but Underreported

We have shown that at least 76.4% of cases occurred in the rainy season from May to December. The average incidence rate of melioidosis in these two provinces was relatively lower than that of the Northeast (2.87 vs. 8.73 per 100,000); however, it was slightly lower than the whole country’s incidence rate (3.95 per 100,000) [3]. Compared with the official report (Report 506) from the national surveillance system [17], we found that the number of cases and deaths from our laboratory-based study was higher than that from the official notification in all seven years. This confirms that melioidosis is underreported in both provinces and perhaps throughout the entire southern Thailand. Of note, a report from Alor Setar in Kedah, a northern state of Malaysia that is 50 km away from its Thailand border adjacent to Songkhla, showed a very high incidence rate of melioidosis at 16.35 per 100,000 [18]. In that study, 145 melioidosis cases were reported during 2005–2008. Our finding agrees with other studies that the majority of melioidosis cases occurred during the wet season months [19,20]; especially in Northeast Thailand, most cases occurred from June to November, the rainy season months. This was also confirmed by Pearson correlation coefficient (r) values that the increased number of melioidosis cases in the rainy season had the strength of a linear association with the amount of rainfall (Figure 2). When the number of cases reported in the following month was used in the calculation, the resulting r value was higher (Figure 2c), suggesting that the rainfall amount had a stronger relationship with new cases in the following month. In addition, it has been reported that the disease severity or death from melioidosis is also correlated with the intensity of rainfall [18,21]. Heavy monsoonal rains and winds may cause a shift towards inhalation of *B. pseudomallei* [21]. 

### 4.2. Death from Melioidosis Is a Pressing Issue Nationwide

Although the incidence rate of melioidosis in these two southern provinces was lower than that of the Northeast, the mortality rate in treated patients was comparable. Our data have shown that deaths from melioidosis occurred rapidly; 56 (30.9%) out of 181 deaths occurred within the first two days, or 102 (56.4%) died within seven days of their admissions. Most fatalities (116 out of 181, or 64.1%) occurred during the rainy season. Based on a recent report, the 30-day mortality rate in melioidosis patients in Thailand was 39% [3], while in the data from northern Malaysia, the mortality was 41.8% during 2005–2011 [22]. In contrast, in a nearby developed country like Singapore, where intensive care therapy for sepsis is available, the death rate from melioidosis was reported at 18.4% and decreased by 12.3% annually during 2003–2014 [23]. Moreover, our data have also shown that the case fatality rates (CFR) were high in the group of patients presenting with pneumonia, urinary tract infection, or bacteremia. These patients were diagnosed with culture-positive results from sputum, urine, or hemoculture. We have noted that when two or more cultures were positive, the CFR was also high. This suggests that the positive results from multiple specimens, along with sputum, blood, or urine, should be considered a risk factor for high CFR. The clinicians should be advised on these types of clinical presentations and culture-positive specimens associated with the high CFR, which therefore require special attention and would have a beneficial impact on mortality.

### 4.3. Risks Associated with Preexisting Conditions

It is well recognized that the most common risk factor predisposing individuals to melioidosis is diabetes mellitus, reviewed in [24]. Other reported risk factors or preexisting conditions associated with melioidosis could include hazardous alcohol use, chronic kidney disease, chronic lung disease, thalassemia, cancer, and glucocorticoid or other immunosuppressive therapy [25]. Similar to other studies, we have found that 59.8% of melioidosis patients had at least one known risk factor, while diabetes mellitus was found in 41.3% of patients. Diabetes mellitus was higher in male (131, 69.7%) than female (57, 30.3%) patients. The mortality rate in patients with diabetes mellitus was also higher in males than females: 47 (35.9%) versus 17 (29.8%), respectively. A previous report from Songklanakarind Hospital, one of the four study hospitals, has shown that 73% of melioidosis patients had underlining diseases, and diabetes mellitus was found in 47% of the patients [13]. On another note, in a recent study based on 7126 melioidosis patients in Thailand, 43% of the patients had diabetes mellitus, followed by hypertension (15%) and chronic kidney disease (11%) [3].

### 4.4. Bacteremia, a Common Clinical Manifestation Leading to Septicemic Melioidosis, and Death

It has been reported that heavy bacteremia (>50 cfu/mL) is common in septicemic melioidosis and is usually fatal [26]. At least 274 (58.1%) melioidosis episodes in this study had bacteremia; 83 (30.3%) of them presented with pneumonia, while 123 (44.9%) died from septicemia (data not shown in Table 3). It has been reported that bacteremia on admission can occur in 40–60% of all patients diagnosed with melioidosis, and septic shock occurs in approximately 20% of all cases [24]. Our data have confirmed that deaths from melioidosis occur rapidly and are likely due to uncontrolled bacteremia and septic shock.

## 5. Conclusions

The laboratory-based retrospective study has confirmed that Songkhla and Phatthalung, the two southern provinces of Thailand, are seasonally endemic to melioidosis. Even though the incidence rate is much lower than that of the Northeast, the mortality rate is comparably high. Similar to other studies in the region, melioidosis patients had underlining diseases, especially diabetes mellitus, which was identified as the major preexisting condition. In addition, bacteremia on admission is common and associated with disease severity and septic shock. The findings from these two provinces may be used to estimate the burden of melioidosis throughout southern Thailand. Based on the current incidence rate, we expect to see 210 to 364 melioidosis cases per year in southern Thailand, where almost 10 million people live. Lastly, we have revealed that the official notifications for cases and deaths from melioidosis in both provinces are seriously underreported. The urgent needs will include improving disease surveillance and the reporting system for melioidosis.

## Figures and Tables

**Figure 1 tropicalmed-08-00286-f001:**
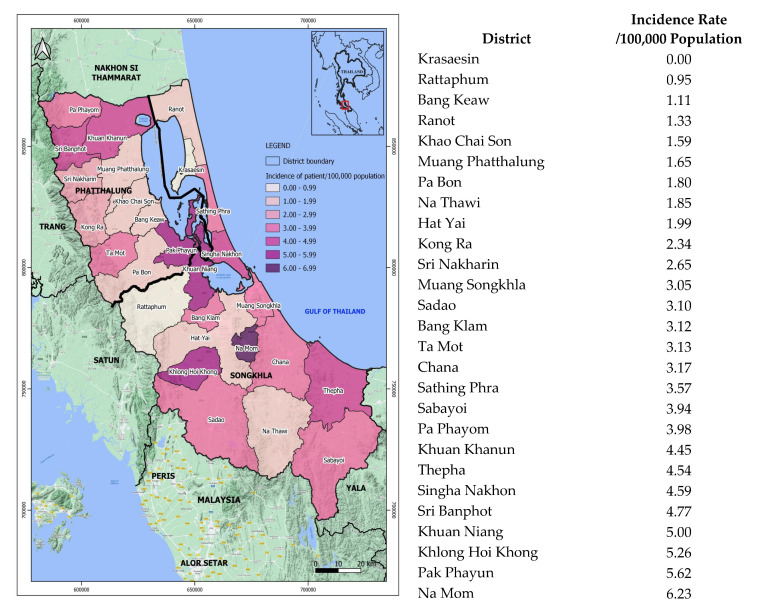
Geographic distribution of melioidosis mapped based on the incidence rates (per 100,000 population) in 27 districts of Songkhla and Phatthalung Provinces, 2014–2020.

**Figure 2 tropicalmed-08-00286-f002:**
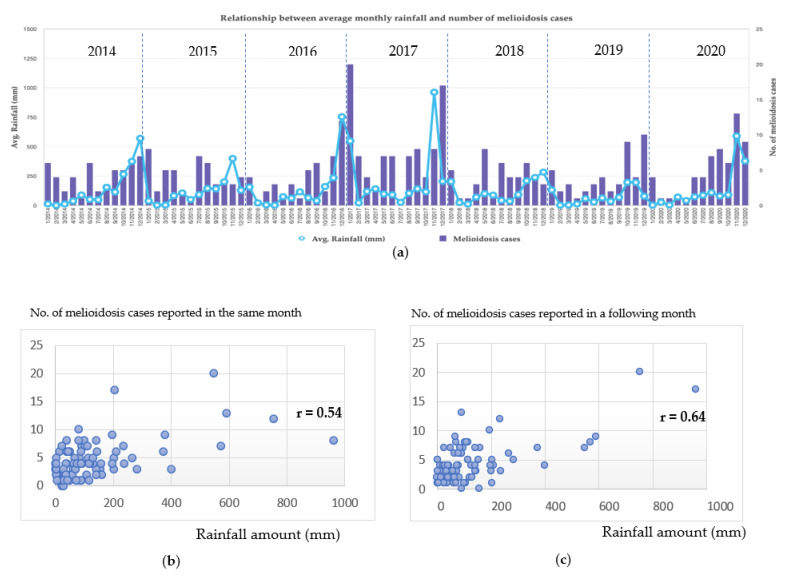
Melioidosis cases strengthened with rainfall; (**a**) showing the average monthly rainfall amount in millimeters (mm) and the number of melioidosis cases; we noted that approximately 76.4% of the infections occurred during the rainy season (May to December), and the increased number of cases in 2016–2017 were associated with two major flooding events that occurred towards the end of both years. Scatterplots show the relationship between the amount of rainfall and the number of cases reported in the same month (**b**) or the number of cases reported one month later (**c**). We noted that the Pearson correlation coefficient (r) in (**c**) was higher than that in (**b**).

**Table 1 tropicalmed-08-00286-t001:** Number of patients with positive cultured specimens.

Positive Specimens	Number of Patients (%)		
	Inpatients		Outpatients (*n* = 18)
	(*n* = 455)	CFR	
Hemoculture	324 (71.2)	45.4%	9 (50.0)
Pus/wound swab	120 (26.4)	19.2%	7 (38.9)
Sputum	104 (22.9)	59.6%	1 (5.6)
Urine	32 (7.0)	46.9%	1 (5.6)
Synovial fluid	20 (4.4)	20.0%	0 (0.0)
Parotid gland abscess	18 (4.0)	5.6%	1 (5.6)
Plural fluid	6 (1.3)	16.7%	0 (0.0)
Spleen/liver abscess	6 (1.3)	33.3%	0 (0.0)
CSF	2 (0.4)	0.0%	0 (0.0)
Positive > 1	117 (25.7)	44.4%	0 (0.0)

**Table 2 tropicalmed-08-00286-t002:** Number of culture-confirmed melioidosis episodes at each study hospital in Songkhla and Phatthalung Provinces from 2014 to 2020.

Province	Hospital Name	2014	2015	2016	2017	2018	2019	2020	Total
Songkhla	Songklanakarind	12	19	17	23	24	13	12	120
	Hatyai	21	22	17	43	15	20	26	164
	Songkhla Provincial	17	14	15	28	14	15	14	117
Phatthalung	Phatthalung Provincial	14	10	11	19	13	11	21	99
Total		64	65	60	113	66	59	73	500

**Table 3 tropicalmed-08-00286-t003:** Demographic and clinical characteristics of patients admitted with culture-confirmed melioidosis at four tertiary care hospitals in Songkhla and Phatthalung Provinces, Thailand, 2014–2020.

Characteristics of Patients	Number of Cases	%
Inpatients	455	96.2
Outpatients	18	3.8
**Sex**		
Male	337	71.2
Female	136	28.8
**Age (years)**		
0–4	12	2.5
5–9	9	1.9
10–14	6	1.3
15–19	3	0.6
20–29	26	5.5
30–39	47	10.0
40–49	86	18.2
50–59	125	26.4
60–69	89	18.8
70–79	44	9.3
>80	26	5.5
Median age: 54 years (IQR 41.5–64)		
**Underlining medical conditions**		
Diabetes mellitus	188	41.3
Cancer/Malignancy	35	7.7
Chronic renal diseases	26	5.7
Immunosuppressant usage	24	5.3
Thalassemia	22	4.8
Excessive alcohol consumption	14	3.1
Chronic lung diseases (e.g., TB)	14	3.1
HBV/HIV	13	2.9
At least one of above	272	59.8
**Clinical Manifestations**	**Number of episodes (%)**	**CFR ^1^**
Bacteremia	274 (58.1%)	45.8%
Pneumonia	166 (35.2%)	56.7%
Skin/wound infection	101 (21.4%)	13.3%
Urinary tract infection	46 (9.7%)	46.7%
Hepatosplenic abscesses	33 (7.0%)	21.9%
Parotid gland infection	18 (3.8%)	5.6%
Septic arthritis	18 (3.8%)	16.7%
CNS infection	2 (0.4%)	0.0%
**Mortality Rate ^1^**	181 (39.8%)	

^1^ Calculated per 455 cases of inpatients.

## Data Availability

Individual patient data will not be available for sharing, but the laboratory data and the culture-confirmed test results can be provided upon request.

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
