# Peer review of "Under-Reporting Cases and Deaths from Melioidosis: A Retrospective Finding in Songkhla and Phatthalung Province of Southern Thailand, 2014–2020"

_tropicalmed, 2023, doi:10.3390/tropicalmed8050286_

Round 1

Reviewer 1 Report

Dear Authors and Editor,

Thank you for considering me for the role of reviewer for this manuscript, which I enjoyed reading. The authors have compiled retrospective data concerning the human disease melioidosis within a region of Thailand, where is previously less well characterised. The data presented show that disease in this region is akin with melioidosis elsewhere in South East Asia. The critical finding is that melioidosis has been under-reported in these regions. We are constantly finding that this disease is more and more widespread and previously though. Instances of undiagnosed melioidosis in an endemic country is further evidence that more funding is needed in this often neglected area.

Generally the use of language is good. I have a number of comments that I feel should be addressed for this manuscript to be published. There is also a question of whether this manuscript fits the scope of the journal, which seems to not include these types of descriptive study and does not list Burkholderia pseudomallei as a NETD (although it clearly is one). This need to be an editorial decision.

The materials and methods should include details regarding how the charts and maps were generated.

The resolution of the figures are too low.

With regards to figure 2, the authors state that rainfall and incidence correlate. This statement would be strengthened should a correlation analysis be performed. The author could include a scatter plot of rainfall (x axis) and incidence (y axis) the value for each month might be plotted. Furthermore a Pearson’s R value for this correlation might be calculated and quoted.

The first statements in the discussion do no wholly follow. The authors write: “We were enthusiastic about the finding from the 7th WMC held in Bangkok, Thailand, in 2013 that suggested two urgent clinical needs for melioidosis; one was to reduce the global incidence of the disease, and another was to improve the clinical outcome of infected patients [14]. Knowing the actual burden of melioidosis in any endemic areas would fulfil these two urgent needs.” These needs not will be fulfilled by “knowing the actual burden of melioidosis”; however knowing the extent of the problem will certainly help in identify the needed scale of the solution.

I think that the concluding remarks should reiterate the difference in government statistics and study-based estimates.

Author Response

Authors’ responses to the reviewer 1’s comments:

Dear Authors and Editor,

Thank you for considering me for the role of reviewer for this manuscript, which I enjoyed reading. The authors have compiled retrospective data concerning the human disease melioidosis within a region of Thailand, where is previously less well characterised. The data presented show that disease in this region is akin with melioidosis elsewhere in South East Asia. The critical finding is that melioidosis has been under-reported in these regions. We are constantly finding that this disease is more and more widespread and previously though. Instances of undiagnosed melioidosis in an endemic country is further evidence that more funding is needed in this often neglected area.

Generally the use of language is good. I have a number of comments that I feel should be addressed for this manuscript to be published. There is also a question of whether this manuscript fits the scope of the journal, which seems to not include these types of descriptive study and does not list Burkholderia pseudomallei as a NETD (although it clearly is one). This need to be an editorial decision.

Authors: Thank you very much for your time reviewing this manuscript. We very much appreciated your comments and suggestions.

Comments and Suggestions for Authors:

  1. The materials and methods should include details regarding how the charts and maps were generated.

Authors: We have provided the method we used to map the incidence rates in more details. 

  1. The resolution of the figures are too low.

Authors: We have increased the resolution for the figures.

  1. With regards to figure 2, the authors state that rainfall and incidence correlate. This statement would be strengthened should a correlation analysis be performed. The author could include a scatter plot of rainfall (x axis) and incidence (y axis) the value for each month might be plotted. Furthermore a Pearson’s R value for this correlation might be calculated and quoted.

Authors: The statement has been corrected. It now reads “Melioidosis cases strengthened with rainfall”. In addition, Pearson correlation coefficient has been added to the analysis. Two scatter plots were made, and the r values were calculated. These plots and the r values have been added to Figure 2. Thank you for the suggestions.

  1. The first statements in the discussion do no wholly follow. The authors write: “We were enthusiastic about the finding from the 7th WMC held in Bangkok, Thailand, in 2013 that suggested two urgent clinical needs for melioidosis; one was to reduce the global incidence of the disease, and another was to improve the clinical outcome of infected patients [14]. Knowing the actual burden of melioidosis in any endemic areas would fulfil these two urgent needs.” These needs not will be fulfilled by “knowing the actual burden of melioidosis”; however knowing the extent of the problem will certainly help in identify the needed scale of the solution.

Authors: We agree with the reviewer’s comments. We have modified the text and the statement now read “Besides finding the actual burden of melioidosis in any endemic areas, knowing the extent of the problems will also help identify the needed scale of the solution to fulfill these two urgent needs”.

  1. I think that the concluding remarks should reiterate the difference in government statistics and study-based estimates.

Authors: We have modified the concluding remarks to reiterate the difference in government statistics and study-based estimates per the reviewer’s suggestion. The last statements now read “Lastly, we have revealed the official notifications for cases and deaths from melioidosis are seriously under-reported. The urgent needs will include improving the disease surveillance and the reporting system for melioidosis” 

Reviewer 2 Report

1. Lines 47-50: Do these data apply for prevalence or incidence?

2. Lines 97-99: please provide a brief introduction about these hospitals for the international reader. How close are they situated? What is the size of their draining population and area? Lines 215-225 should answer this, and can be placed here.

3. Lines 110-111: Should you add "immunosuppressant use" here? (See Line 162.)

4. Line 215: "Enthusiastic about" or "enthused by"?

5. This paper conclusively shows that melioidosis is under-reported in this part of Thailand. However, another objective has been to reduce the number of deaths. Melioidosis is known to have several clinical presentations ranging from acute septicaemic to chronic, and it is those cases closer to the left of this spectrum that have a high case fatality rate (CFR). Therefore, it would have been more useful if you had analysed he data in relation to the clinical presentation. The presentation could have been determined both from the case records and the types of specimens from the which the organism was grown. By doing this, I think an even higher CFR for acute presentations would have been obtained. Also, clinicians could be advised on the types of presentations associated with the high CFR and therefore requiring special attention, which would have a beneficial impact on mortality.

6. The tile should be changed to reflect the main argument: under-reporting of cases and deaths. I am not sure if this methodology can support the term 'surveillance', because this is retrospective. 

One correction suggested.

Author Response

Authors’ responses to the reviewer 2’s comments:

Authors: Thank you very much for your time reviewing this manuscript. We very much appreciated your comments and suggestions.

Comments and Suggestions for Authors:

  1. Lines 47-50: Do these data apply for prevalence or incidence?

Authors: Corrected. The sentence now reads “The most incidences were from the Northeast with 5,475 cases, while the most undersized majority was from the West with only 19 cases”.

  1. Lines 97-99: please provide a brief introduction about these hospitals for the international reader. How close are they situated? What is the size of their draining population and area? Lines 215-225 should answer this, and can be placed here.

Authors: We have added information about these hospitals e.g., sizes and locations to the manuscript per the reviewer’s suggestion.

  1. Lines 110-111: Should you add "immunosuppressant use" here? (See Line 162.)

Authors: Added. Thank you.

  1. Line 215: "Enthusiastic about" or "enthused by"?

Authors: Corrected. Yes, we were enthused by the finding from the 7th WMC.

  1. This paper conclusively shows that melioidosis is under-reported in this part of Thailand. However, another objective has been to reduce the number of deaths. Melioidosis is known to have several clinical presentations ranging from acute septicaemic to chronic, and it is those cases closer to the left of this spectrum that have a high case fatality rate (CFR). Therefore, it would have been more useful if you had analysed he data in relation to the clinical presentation. The presentation could have been determined both from the case records and the types of specimens from the which the organism was grown. By doing this, I think an even higher CFR for acute presentations would have been obtained. Also, clinicians could be advised on the types of presentations associated with the high CFR and therefore requiring special attention, which would have a beneficial impact on mortality.

Authors: This is a great suggestion. We have calculated the CFR for these acute presentations and their associated culture results. The CFRs have been added to Table 1 and 2, and further discussed in the revised manuscript per the reviewer’s suggestion.   

  1. The tile should be changed to reflect the main argument: under-reporting of cases and deaths. I am not sure if this methodology can support the term 'surveillance', because this is retrospective. 

Authors: Corrected. We have changed the manuscript’s title to “Under-reporting cases and deaths from melioidosis: a retrospective study in Songkhla and Phatthalung Province of southern Thailand, 2014-2020.

  1. Comments on the Quality of English Language: One correction suggested.

Authors: Thank you for correcting the wording.

Round 2

Reviewer 2 Report

The revised manuscript is very good.

I noted that the CFR is high also when two (or more) cultures are positive. Should you add this also as a risk factor for high CFR (along with sputum/blood/urine culture positivity)? This is the only new suggestion I have.

Author Response

We very much appreciated the last comments/thoughts from the reviewer 2. We then incorporated that statement into the manuscript.  It now reads "We have noted that when two or more cultures were positive, the CFR was also high. This suggests that the positive results from multiple specimens along with sputum, blood or urine, should be considered as a risk factor for high CFR". 

Thank you very much.